# A scoping review on the use of traditional medicine and oral health in Africa

**Moréniké Oluwátóyìn Foláyan** [1,2]*, **Mobolaji Timothy Olagunju** [3], **Olunike Rebecca Abodunrin** [4], **Omolola Titilayo Alade** [1,5]

1 Oral Health Initiative, Nigerian Institute of Medical Research, Yaba, Lagos, Nigeria, 2 Department of Child Dental Health, Obafemi Awolowo University, Ile-Ife, Nigeria, 3 Department of Epidemiology and Biostatistics, Nanjing Medical University, Nanjing, Jiangsu Province, China, 4 Lagos State Health Management Health Agency, Lagos, Lagos State, Nigeria, 5 Department of Preventive and Community Dentistry, Obafemi Awolowo University, Ile-Ife, Nigeria

* toyinukpong@yahoo.co.uk

**Data Availability Statement:** All relevant data are within the manuscript and its Supporting Information files.

**Funding:** The author(s) received no specific funding for this work.

## Abstract

### Background

This review aimed to chart the landscape of literature concerning the precise applications of traditional medicine in managing specific oral diseases and, in doing so, to pinpoint knowledge gaps surrounding the use of traditional medicine for oral disease management in the African context.

### Methods

A systematic search of the literature was conducted on PubMed, Web of Science, Scopus, and CINAHL. The search was conducted from the inception of the database till September 2023. A search of related citations and references was also carried out. Only English language publications were included. A summary of studies that met the inclusion criteria was conducted.

### Results

Of the 584 records identified, 11 were duplicates and 12 studies, published between 2006 and 2021, met the inclusion criteria. The studies were published from eight countries located in the five sub-regions on the continent. All the studies were either experimental designs or ethnobotanical surveys and they all utilized plant-based remedies. The five experimental studies aimed to assess the impact of whole plants or plant extracts on the three microorganisms responsible for dental caries and seven responsible for periodontal diseases. The number of plant species identified by the seven ethnobotanical surveys ranged from 29 to 62 while the number of plan families ranged from 15 to 29. The remedies were either topical applied, use as mouth rinses, gargled, or chewed. The systemic routes of administration identified were inhalation and drinking. The remedies were used for the treatment of hard such as dental caries and tooth sensitivity, to soft tissue lesions such as mouth ulcers, gingival bleeding, and mouth thrush. Other oral disorders managed include halitosis, jaw fracture, and oral cancer.

**Competing interests:** The authors have declared that no competing interests exist.

## Conclusions

Given the increasing prevalence of oral diseases within the region, the shortage of oral healthcare professionals and limited access to financial resources, it becomes imperative to support the generation of empirical evidence to enhance the provision of traditional medicine for oral healthcare in Africa.

## Introduction

Traditional medicine encompasses a comprehensive body of knowledge, skills, and practices rooted in the beliefs, experiences, and customs of various cultures [1]. It serves multiple purposes, including the preservation of health, the prevention of ailments, the diagnosis of conditions, and the treatment of physical and mental illnesses [2]. Traditional medicine often employs natural resources, such as plants, animals, fungi, and even geological elements like rocks and minerals, regardless of whether their mechanisms are scientifically understood [3]. This approach is deeply entwined with indigenous beliefs, practical expertise, and ancestral wisdom passed down through generations [4].

Traditional medicine is the primary and sometimes sole recourse for disease prevention and healing in some regions characterized by exclusion and severe poverty and limited access to conventional healthcare services [5]. It plays a vital role in the healthcare landscape of low-income countries, where its utilization is widespread: about 40% to 71% of the population in these regions use traditional medicine [6], and in sub-Saharan Africa, the average prevalence rate of use is 58.2% [7].

While the safe and appropriate use of traditional medicine can offer benefits [8], its concurrent use alongside conventional medical treatments raises concerns regarding its clinical implications [9]. Natural resources and traditional knowledge have played a pivotal role in the development of approximately 40% of pharmaceutical products [10]. As a result, there is a growing interest in exploring the wisdom of ancient cultures, traditional community-based healthcare practices, and natural resources to address contemporary health challenges [ditto].

Traditional medicine is frequently employed in the context of oral healthcare to alleviate toothache, address periodontal inflammation, and manage oral mucosal diseases [11, 12]. These practices encompass the utilization of medicinal plants for oral hygiene, as well as for their analgesic, antibacterial, antifungal, antiseptic, antimicrobial, antioxidant, and antiviral properties [13]. Nevertheless, there remains a limited understanding of the specific types of traditional medicine used for the management of distinct oral diseases, particularly in Africa where such practices are widespread.

The primary objective of this scoping review is to comprehensively examine the existing body of literature concerning the utilization of traditional medicine for oral healthcare in Africa. This review aims to chart the landscape of literature concerning the precise applications of traditional medicine in managing specific oral diseases and, in doing so, to pinpoint knowledge gaps surrounding the use of traditional medicine for oral disease management in the African context.

## Methods

The methodology was based on the Joanna Briggs Institute (JBI) scoping review methodology [14] and reported following the Preferred Reporting Items for Systematic Reviews and Meta-Analyses Extension for Scoping Reviews guidelines (PRISMA-ScR) [15, 16].

### Research question

The review was guided by the research question: what are published about the use of traditional medicines for oral health in Africa?

### Identifying relevant studies

A systematic search of the literature was conducted on PubMed, Web of Science, Scopus, and the Cumulative Index of Nursing and Allied Health Literature (CINAHL) using the terms shown in S1 File. The search was conducted from the inception of the database till September 2023. A search of related citations and references was also carried out. No author was contacted.

### Study selection

Publications identified through the search strategy were downloaded into Endnote, and imported into Rayyan, and duplicates were removed. Three researchers (ORA, MTO, OTA) performed a screening of the titles and abstracts of the downloaded articles independently using pre-defined inclusion and exclusion criteria. Studies were included if there was an agreement between all the reviewers. Where there were disparities, a fourth reviewer (MOF) was consulted to sort out any disparity between the three reviewers. In addition, three researchers completed the full-text review (ORA, MTO, OTA). Uncertainty regarding whether publications met the inclusion criteria was resolved via consensus among the three researchers or with recourse to the fourth reviewer (MOF).

### Inclusion criteria

Peer-reviewed journal articles, books, book chapters, conference proceedings, and reports that cover traditional medicine and oral health focusing on Africa that addressed both preventive and curative aspects herbal remedies and oral health were included in the review. Studies reporting outcomes related to the impact of traditional medicine on oral health, including oral diseases, oral hygiene, and oral health-related quality of life were also included. There was no restriction on study design or date of publication. There was, however, restriction on language with only studies published in English Language included in this scoping review.

### Exclusion criteria

Animal studies or in vitro experiments were excluded. Also excluded unpublished theses and dissertations, letters to the editor, commentaries on studies, scoping, systematic and narrative reviews, and studies whose full lengths cannot be accessed. In addition, studies with insufficient data or methodology not suitable for analysis, and those published on websites were also excluded.

### Data charting process

A data-charting form was developed to extract relevant variables. The charted variables were the literature characteristics (e.g., authors, year of publication), study aim, study design, form of traditional medicines used, oral diseases managed, and the outcomes of the studies.

### Data analysis

The results of the scoping review were reported according to the PRISMA-ScR checklist. A deductive analysis was conducted using the framework developed for the data extraction.

Details generated from the analysis were the years of publication, regions in Africa (Western, Eastern, Northern, Central and Southern Africa) [https://www.uneca.org/subregional-offices] where studies were conducted, forms of traditional medicines used (plants, plants, animals, fungi, or geological elements), oral diseases (and oral diseases pathogenic organisms) investigated, plants (and plant extracts) investigated, and forms and routes of administration of the traditional medicines for the management of oral diseases.

## Results

The search resulted in the identification of 584 records, which were downloaded into Endnote 7.8 and imported into Rayyan. After de-duplication, 562 records remained. After reviewing titles and abstracts, screening, 26 articles were eligible for full-text screening. On screening the full articles, 13 articles were excluded either because full articles were not available, the focus of the study was not Africa or the study was not related to oral health, leaving 13 articles [17–29] for this review. Fig 1 shows the flow diagram of the publication screening process.

### Study characteristics

Table 1 presents an overview of the 12 studies included in this analysis. These studies were conducted over a span of years, ranging from 2006 to 2021. Specifically, three of them were carried out between 2000 and 2010 [25, 27, 28], eight studies were conducted between 2011 and 2020 [17–20, 22–24, 26], and one study was conducted in 2021 [21].

Regarding the geographic distribution of these studies, a quarter of them (25.0%) were conducted in Southern Africa, specifically in South Africa [17, 26, 27]. East Africa was represented by three studies (23.1%): two from Uganda [21, 22] and one from Ethiopia [20]. North Africa accounted for three studies (23.1%): two from Morocco [23, 24] and one from Sudan [19]. West Africa contributed two studies (15.4%): one from Nigeria [25] and one from Burkina Faso [28]. Additionally, one study (8.3%) was conducted in Central Africa, specifically in Cameroon [18].

### Study designs

All the studies followed either experimental designs [17, 19, 21, 22, 27] or ethnobotanical surveys [18, 20, 23–26, 28]. All these investigations focused on traditional oral health management practices that primarily utilized plant-based remedies. There was one exception, where minerals were used in combination with plant-based treatments [18].

Table 1 highlights the traditional medicines used to manage various oral diseases and conditions. These range tooth related diseases: toothache [18, 23–26, 28], dental caries [18, 19, 23, 24, 26], tooth bleaching [18, 23, 24], broken teeth [18], and tooth sensitivity [18, 24]; to soft tissue diseases: mouth sores and ulcers [18, 23–26], abscesses [18, 23, 24, 26], gingivitis and gingival bleeding [18, 23–26, 28], oral infections such as syphilis [18, 23], and mouth thrush [18]. Other oral disorders include halitosis [18, 23, 24, 26], broken jaw [18], temporomandibular joint pain [18], oral cancer [18], tooth extraction [18], and tooth strengthening [26]. There were also traditional medicinal products for oral care [23–25].

### Characteristics of experimental studies

Table 2 shows the characteristics of the experimental studies. The experimental studies aimed to assess the impact of whole plants or plant extracts on the microorganisms responsible for dental caries [17, 19, 21, 22, 27] and periodontal diseases [17, 22, 27]. The microorganisms targeted for caries studies included *Streptococcus mutans* [17, 21, 22, 27], *Streptococcus sobrinus*

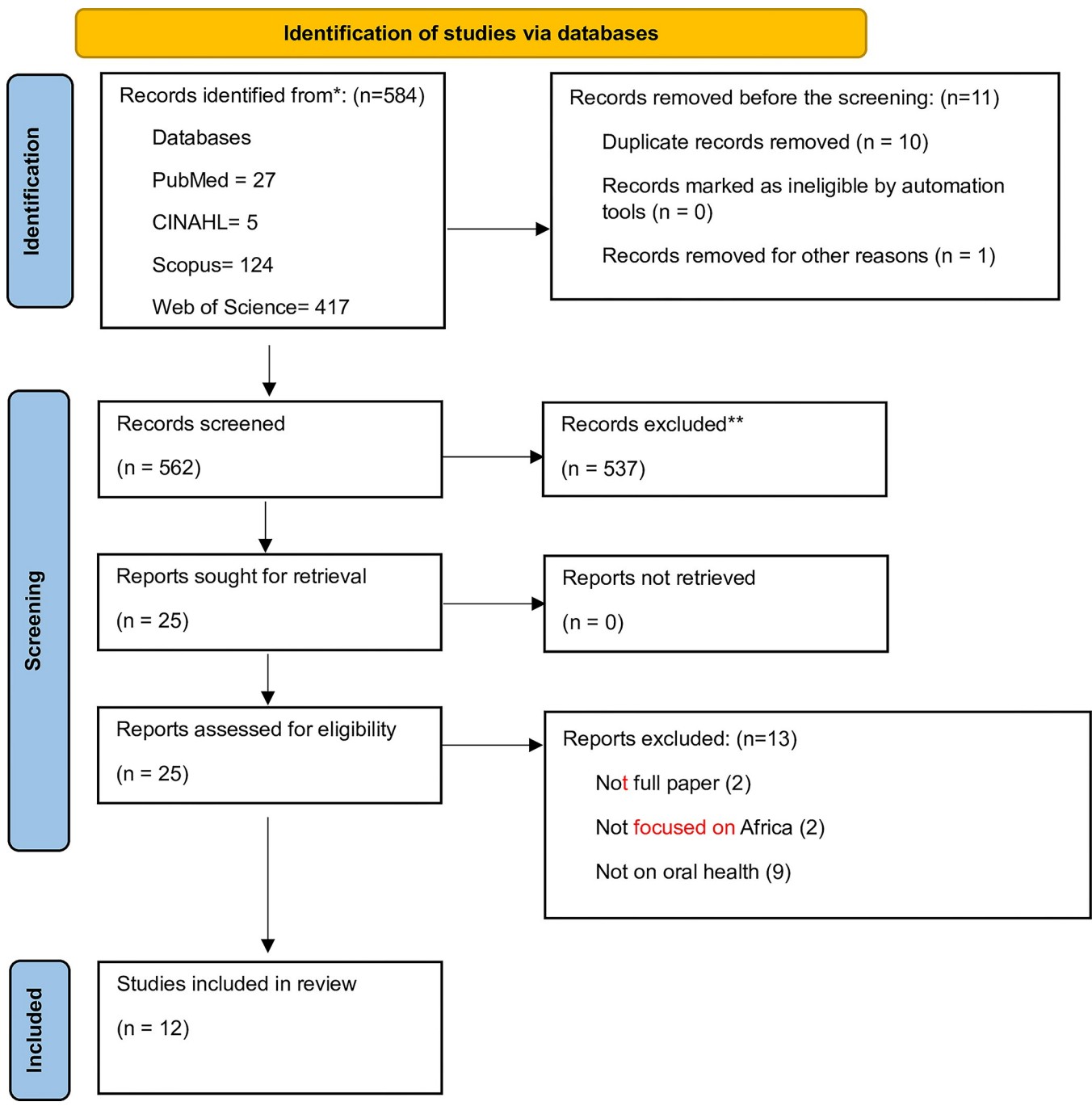

**Fig 1. Study flowchart showing the flow of studies from retrieval to the final included studies.**

[19, 21], and *Lactobacillus acidophilus* [21, 22]. In periodontopathic organisms studied were *Fusobacterium nucleatum* [17], *Porphyromonas gingivalis* [22, 27], *Aggregatibacter actinomyce-temcomitans* [22, 27], *Actinomyces naeslundii* [27], *Actinomyces israelii* [27], *Candida albicans* [27], and *Prevotella intermedia* [27]. All the studies reported on the outcome of the activities of the plants on the pathogenic organisms.

**Table 1. Characteristics of included studies.**

| S/no | Title | Author (Year of publication) | Country | Study Aim | Study design |
|---|---|---|---|---|---|
| 1 | Antimicrobial activity and toxicity of extracts from the bark and leaves of South African indigenous Meliaceae against selected pathogens | Oyedeji-Amusa et al., (2020) [17] | South Africa | The study evaluated the antimicrobial efficacy and toxicity of aqueous, methanolic and dichloromethane leaf and bark extracts of South African Meliaceae against selected pathogens. *Ocimum basilicum* | Experimental |
| 2 | Ethnomedicinal plants used by traditional healers to treat oral health problems in Cameroon | Agbor and Naidoo (2015) [18] | Cameroon | To determine the therapeutic methods used by traditional healers to treat oral diseases in Cameroon. | Ethnobotanical survey |
| 3 | Anti-caries activity of selected Sudanese Emphasis medicinal plants with on *Terminalia laxiflora* | Mohieldin et al. (2017) [19] | Sudan | To determine whether *Terminalia laxiflora* methanolic wood extracts can be used to prevent or treat oral health problems. | Experimental |
| 4 | Medicinal plants and their uses by the people in the Region of Randa, Djibouti | Hassan-Abdallah et al. (2013) [20] | Ethiopia | To assess plant-related ethnomedinal knowledge of the people in Randa Region | Ethnobotanical survey |
| 5 | Antibacterial activities of extracts from Ugandan medicinal plants used for oral care | Ocheng et al. (2014) [21] | Uganda | To investigate antibacterial activities of 16 commonly used medicinal plants on microorganisms associated with periodontal diseases and dental caries. | Experimental |
| 6 | Essential oils from Ugandan aromatic medicinal plants: chemical composition and growth inhibitory effects on oral pathogens | Ocheng et al., (2015) [22] | Uganda | Assessed the growth inhibitory effects of essential oils extracted from ten Ugandan medicinal plants used traditionally in the management of oral diseases against oral pathogens. | Experimental |
| 7 | Oral disorders and ethnobotanical treatments: A field study in the central Middle Atlas (Morocco) | Najem et al. (2020) [23] | Morocco | An inventory of the plants used in the treatment of oral diseases, as well as to document possible risks of intoxication. | Ethnobotanical survey |
| 8 | Medicinal and aromatic plants used in traditional treatment of the oral pathology: the ethnobotanical survey in the economic Capital Casablanca, Morocco (north Africa) | Zougagh et al. (2019) [24] | Morocco | To identify the medicinal and aromatic plants most requested for the treatment of the most common oral pathology | Ethnobotanical survey |
| 9 | Ethnobotanical plants used for oral healthcare among the Esan tribe of Edo State, Nigeria. | Idu et al. (2009) [25] | Nigeria | To enumerate some plant species and their indigenous uses as chewing sticks for dental and oral healthcare, by the locals of some communities in Esan region of Edo State, Nigeria. | Ethnobotanical survey |
| 10 | Medicinal plants used to treat oral diseases in the Lepelle-Nkumpi Municipality, Limpopo Province, South Africa | Semenya (2019) [26] | South Africa | To document the medicinal plants used to treat oral diseases in the Lepelle-Nkumpi Municipality in the Limpopo Province, South Africa. | Ethnobotanical survey |
| 11 | Antimicrobial activity of medicinal plants against oral microorganisms | More (2008) [27] | South Africa | To determine the antimicrobial activity of the traditional South African medicinal plants used as chewing sticks against oral microorganisms which are responsible for dental caries | Experimental |
| 12 | Use of medicinal plants for the treatment of oral diseases in Burkina Faso | Tapsoba and Deschamps. (2006) [28] | Burkina Faso | To document the use of medicinal plants in the treatment of oral diseases in the Kadiogo province of Burkina Faso. | Ethnobotanical survey |

## Characteristics of ethnobotanical surveys

Table 3 delves into the diverse array of plant-based products used in the management of oral diseases. The number of plant species investigated per study ranged from 29 to 62 different species. The plant families studied per study ranged from 15 to 29 different families. Plant families prominently used in the management of oral diseases were: Solanaceae [20, 26, 28], Aristolochiaceae [20, 26], Asteraceae [23, 24], Lamiaceae [23, 24], Apiaceae [23, 24], Myrtaceae [24, 28], Rosaceae [23], Anacardiaceae [25], Fabaceae [25], Rutaceae [25], Euphorbiaceae [25], Rhamnaceae [28], Convolvulaceae [28], Caricaceae [28], Verbenaceae [28], Sterculiaceae [28], Scrophulariaceae [28], Rubiaceae [28], Polygalaceae [28], and Poaceae [28].

These traditional medicinal products are derived from various parts of the plants, namely: roots [18, 23–26, 28], leaves [18, 23, 24, 26, 28], stems, twigs, and branches [18, 23–25, 28], barks [23–25, 28], fruits [18, 23, 24], whole plants [18, 24], flowers [23, 24], rhizomes [24, 26],

**Table 2. Characteristics of the experimental studies included in the scoping review.**

| S/no | Author (Year of publication) | Country | Form of traditional medicine (no. of products studied) | Oral diseases explored | Study findings |
|---|---|---|---|---|---|
| 1 | Oyedeji-Amusa et al., (2020) [17] | South Africa | Plant leaf and bark solvent extracts | Dental caries (*Streptococcus mutans*) Periodontal diseases (*Fusobacterium nucleatum* sub *nucleatum*) | The DCM leaf and bark extracts and methanol leaf extracts of the eight samples showed moderate activity against the 2 pathogens. The methanol bark extract and aqueous extracts showed activities against *S. mutans*. |
| 2 | Mohieldin et al. (2017) [19] | Sudan | Plant bark, leaves, wood, and stem | Caries: *S. sobrinus* and glucosyltransferase inhibitory effects | Methanolic extracts of *Terminalia brownii* (bark), *T. laxiflora* (wood), *A. seyal* (bark), Persicaria *glabra* (leaves) and *Tamarix nilotica* (stem) showed good activities against both *S. sobrinus* and glucosyltransferase. |
| 3 | Ocheng et al. (2014) [21] | Uganda | Pulp juice and solvent extracts (hexane, methanol and water) from leaves and aerial part. | Dental caries (*Streptococcus mutans*, *Streptococcus sobrinus*, *Lactobacillus acidophilus*) | Pulp juice of *Zanthoxylum chalybeum* and *Euclea latidens* showed activity against all the bacteria, Zanthoxylum chalybeum being most active. Hexane extract from aerial part of Helichrysum *odoratissimum* was most active. Methanol extract from leaves of Lantana *trifolia* showed activity against all bacteria |
| 4 | Ocheng et al., (2015) [22] | Uganda | Oil extracts | Periodontal disease (*Porphyromonas gingivalis and Aggregatibacter actinomycetemcomitans* Caries (*Streptococcus mutans and Lactobacillus acidophilus*) | Essential oils from the studied plants show marked growth inhibitory effects on periodontopathic *Aggregatibacter actinomycetemcomitans* and *Porphyromonas gingivalis*, moderate effects on cariogenic *Streptococcus mutans*, and the least effect on *Lactobacillus acidophilus*. |
| 5 | More (2008) [27] | South Africa | Ethanol extracts of eight plants | *Actinobacillus actinomycetemcomitans, Actinomyces naeslundii, Actinomyces israelii, Candida albicans, Porphyromonus gingivalis, Privotella intermedia and Streptococcus mutans* | Six plants exhibited MIC values ranging from 25.0 mg/ml to 0.8 mg/ml. Gram negative bacteria were found to be more resistant to the plant extracts than Gram positive bacteria, except for *Euclea natalensis* which inhibited all three Gram negative bacteria tested in this study |

resins [23], umbels [23], seeds [23, 24, 26], saps [23], cloves [24], buds [24], trunks [24], stigmas [24], and bulbs [26].

The modes of administration for these traditional medicinal products encompass a range of practices, namely: topical application of leaves, saps, pastes, or powders [18, 20, 23, 24, 26], mouth rinses [18, 20, 23, 24, 26, 28], gargling [18, 23, 28], brushing [18, 23, 24] and chewing [18, 23, 25, 26]. There were two systemic routes of administration identified: inhalation [23, 26, 28], and drinking [28].

## Discussion

This scoping review provides an overview of traditional medicine practices in oral healthcare across Africa. It encompasses a comprehensive summary of the various types of traditional remedies, their forms, routes of administration, and includes limited insights into experimental studies that assess the properties of plant extracts in combatting pathogenic organisms associated with dental caries and periodontal diseases. The study findings indicate that publications on the use of traditional medicine for the management of oral diseases were from the five sub-regions in Africa, all the studies used plant-based products and the studies were majorly ethnobotanical surveys. All the experimental studies reported on the effects of the plants and plant extracts on the cariogenic or periopathogenic organisms studied. The ethnobotanical studies also indicated the use of plant-based products to manage caries, caries related complications and periodontal diseases in addition to managing other oral diseases such as halitosis, fractured teeth and jaw bones, and the management of oral cancers. The traditional medicine products were majorly used topically through topical applications or mouth rinsing/gargling. There were two reports on systemic applications–ingestion and inhalation.

**Table 3. Characteristics of the ethnobotanical surveys included in the scoping review.**

| S/no | Author (Year of publication) | Country | Forms in which products are used | Form of traditional medicine | Oral diseases studied | Predominant family of plants used for oral diseases |
|---|---|---|---|---|---|---|
| 1 | Agbor and Naidoo (2015) [18] | Cameroon | Topical application of paste, mouth rinses, gargling toothbrushing, directing the smoke from melted paste into the mouth, chewing. | Plant roots, leaf, bark, seeds, stems, fruits, whole plants, and minerals as adjunct—sulphur, alum, bicarbonate, calcium carbonate | Toothache, sore mouth, dry mouth, abscess, broken tooth and jaw, tooth sensitivity, mouth thrush, dental caries, gingivitis, oral syphilis, oral cancer, TMJ pain, halitosis, tooth bleaching and dental extraction | 52 plants were studied. Information not available |
| 2 | Hassan-Abdallah et al. (2013) [20] | Ethiopia | Mouth rinse, topical application of paste | Leaf (as mouth wash or paste) | Dental caries | Aristolochiaceae and Solanaceae were the predominant species used. Informant consensus factor was high for use of some species for mouth diseases. |
| 3 | Najem et al. (2020) [23] | Morocco | Chewing, gargling, topical application of powder or sap, mouth rinse, direct application of leaves, brushing, inhalation, | Resin, flowers, umbels, fruit, seeds, leaves, root, bark, sap, branches | Bad breath, dental caries, gingivitis, dental abscesses, toothache, gingival bleeding, mouth ulcers, herpes labialis, stomatitis, tooth whitening, mouth infection, scurvy, oral care | 29 plants from 15 families were studied. The Asteraceae, Lamiaceae, Apiaceae and the Rosaceae were the predominant species used. |
| 4 | Zougagh et al. (2019) [24] | Morocco | Direct application, mouth wash, brushing. | Clove, leaves, flowers, seeds, root, bud, bark of root, trunk, stem or fruit, Rhizome, whole plant, stigma | Gingivitis, periodontal abscess, toothache, tooth sensitivity, halitosis, aphtous, ulcers and herpes, teeth cleaning, tooth sparkling, tooth bleaching, caries | 46 plants from 22 families. Lamiaceae, Apiaceae, Asteracea and Myrtaceae were the predominant species used. The medicinal ones were dominated by Lamiaceae and Asteracea. |
| 5 | Idu et al. (2009) [25] | Nigeria | Chewing | Twigs, root, stem bark, stem, bark, root. | Tooth ache, tooth cleaning, sore gum, gingivitis | 32 plants from 23 families were studied. Anacardiaceae, Fabaceae, Rutaceae, and the Euphorbiaceae were the predominant species used. |
| 6 | Semenya (2019) [26] | South Africa | Chewing, mouthwash, topically apply extract, inhale, crush leave missed with salt topically applied. | Leaves, roots, seeds, bulb, rhizomes | Bad breath, toothache, dental caries, bleeding gum, dental abscesses, mouth ulcer, swollen gums, strengthen the tooth | 41 plants from 30 families. The most predominant species reported as remedies for different oral diseases were the Asteraceae and Solanaceae. |
| 7 | Tapsoba and Deschamps. (2006) [28] | Burkina Faso | Drinking, mouth-washing, gargling or inhalation | Fresh or dried roots, stems, leaves, and bark | Toothache, gingivitis, bleeding | 62 plants from 29 families. The most predominant species reported as remedies for oral diseases were the Rhamnaceae, Convolvulaceae, Myrtaceae, Caricaceae, Verbenaceae, Sterculiaceae, Solanaceae, Scrophulariaceae, Rubiaceae, Polygalaceae and Poaceae. |

One of the strengths of this study is its comprehensive overview of research on traditional medicine employed for oral disease management in Africa, a continent known for its significant reliance on traditional healthcare practices for its population's well-being [29]. However, it is important to acknowledge the study's limitations. It primarily focuses on publications available in English and accessible through the specific database utilized for this research. It's worth noting that research publications in Africa encompass a variety of languages, including French and German [29]. Consequently, the inclusion criteria applied in this scoping review may have unintentionally excluded relevant publications on this subject. Despite this limitation, the current study offers valuable insights into the topic at hand.

Firstly, despite the inclusion of 13 publications stemming from research conducted across the five sub-regions of Africa, this number remains disproportionately low for a continent that heavily relies on traditional healthcare practices. These publications are distributed across only eight (17.1%) out of the 56 countries on the continent. It is essential to emphasize the necessity for country-specific publications since traditional medicine is deeply embedded in the unique cultural contexts of each nation [30]. Moreover, traditional offers a diverse array of health benefits, encompassing both chronic and non-communicable diseases, including those related to oral health [31].

The prevalence of oral diseases is on the rise in Africa and marginalized people are worse affected [32]. In addition, the healthcare resources to address this growing challenge remain scarce across the continent [33]. Consequently, traditional medicines are likely to continue playing a pivotal role in oral disease management in the region for the foreseeable future. To catalyze meaningful improvements in population health, the publication of research findings is imperative. Although there is a noticeable global trend towards increased research publications on traditional medicines, our findings indicate that the rate of growth in publications related to traditional medicine and oral health in Africa does not match this global pace. In addition, the focus has been less on oral health past analysis of publications on traditional medicines from countries in Africa had focused on other diseases like Alzheimer's [34], hypertension, HIV, tuberculosis, malaria, and bleeding disorders [35]. The current study highlights the paramount need to direct attention and provide support to address research on the link between traditional medicines and oral health as a pressing research need area. Doing so can facilitate the development of evidence-based policies that hold the potential to significantly enhance the oral healthcare landscape of countries in the region.

Secondly, the ethnobiological investigations shed light on promising prospects for aligning specific plant families with the management of oral diseases. Our analysis unveiled approximately 20 noteworthy plant families that have been recognized for their potential in oral disease management. The most prominent plant family identified for the management of oral diseases was Solanaceae. It has a widespread distribution but particularly concentrated in South America [36]. The family's most economically significant genus in Africa is Solanum notably *Solanum tuberosum* (potato), *Solanum lycopersicum* (tomato), and *Solanum melongena* (eggplant) [37]. In both rural and urban India, various parts of 17 Solanaceae species, such as fruits, seeds, berries, flowers, twigs, leaves, and roots, as well as the whole plant, have been traditionally employed for dental care to manage toothache [38]. Solanaceae is rich in biologically active alkaloids and exhibits antioxidant and anti-inflammatory properties [39]. When utilized as an adjunctive antiseptic mouth rinse by individuals adhering to recommended oral hygiene practices like brushing and flossing, it effectively reduces the risk of periodontal diseases [40].

Solanaceae and several other plant families are of importance for oral health because of their antimicrobial properties [41]. Studies on the use of these natural plants for oral health are being conducted in other countries outside of Africa such as Australia, Brazil, France, Iran, Japan, Mexico, Switzerland, and the United Kingdom. However, there is a notable scarcity of studies examining the clinical efficacy and toxicity of these products for oral health within Africa: several plants recommended for oral disease treatment may possess inherent dangers and toxicity [23]. In contrast to other regions globally where research explores incorporating these natural products into toothpaste formulations, African countries currently prioritize bioprospecting and primary production of the raw plants known for their antimicrobial properties [41]. To address this concern and pave the way for informed recommendations, further comprehensive studies are imperative. These studies should delve into both the safety and efficacy of traditional medicinal practices employed for oral disease management. By conducting

rigorous research, we can ascertain which traditional remedies are both safe and effective for managing specific oral diseases, enabling the formulation of evidence-based guidelines to promote their usage.

Finally, traditional medicine could play a pivotal role in addressing healthcare disparities. Traditional remedies often come at a significantly reduced cost, sometimes up to 50% less expensive than their conventional counterparts. Additionally, the payment arrangements for traditional medicine can be more flexible, with options such as credit or even labor-based compensation. Furthermore, payment for these services may be contingent upon their success [42]. These accessible and adaptable packages offered by traditional medicine providers can make these treatments more appealing, particularly to individuals facing financial hardship. By enhancing the practice of traditional medicine in the realm of oral healthcare, we have the potential to tackle the pressing issues of unequal access to oral health services in low-income African countries [43]. This, in turn, can contribute significantly to achieving the Sustainable Development Goals, particularly Sustainable Development Goal 3, which emphasizes ensuring health and well-being for all, regardless of age or socioeconomic status [44].

## Conclusions

This scoping review reveals that there exists a body of literature concerning the utilization of traditional medicine in the management of oral diseases across all the sub-regions of Africa. However, these publications are relatively scarce and originate from less than one-fifth of the countries on the continent. Given the increasing prevalence of oral diseases within the region, coupled with a shortage of oral healthcare professionals and limited access to financial resources, it becomes imperative to provide support for the generation of empirical evidence to enhance the provision of traditional medicine for oral healthcare in Africa. Investments are urgently required to facilitate research on traditional medicine's role in oral healthcare, as well as the translation of research findings into practical applications. These endeavors hold the potential to make a significant contribution towards the achievement of Sustainable Development Goal 3, which aims to ensure good health and well-being for all.

## Supporting information

**S1 Checklist. Preferred Reporting Items for Systematic reviews and Meta-Analyses extension for Scoping Reviews (PRISMA-ScR) checklist.**
(DOCX)

**S1 File. Search strategy for the four databases used for the scoping review.**
(DOCX)

## Author Contributions

**Conceptualization:** Moréniké Oluwátóyìn Foláyan.

**Data curation:** Moréniké Oluwátóyìn Foláyan, Mobolaji Timothy Olagunju, Olunike Rebecca Abodunrin, Omolola Titilayo Alade.

**Formal analysis:** Moréniké Oluwátóyìn Foláyan.

**Methodology:** Moréniké Oluwátóyìn Foláyan, Mobolaji Timothy Olagunju, Olunike Rebecca Abodunrin, Omolola Titilayo Alade.

**Project administration:** Moréniké Oluwátóyìn Foláyan.

**Supervision:** Moréniké Oluwátóyìn Foláyan.

**Visualization:** Mobolaji Timothy Olagunju.

**Writing – original draft:** Móréníké Oluwátóyìn Foláyan.

**Writing – review & editing:** Móréníké Oluwátóyìn Foláyan, Mobolaji Timothy Olagunju, Olunike Rebecca Abodunrin, Omolola Titilayo Alade.

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
