## [Decision Letter · Decision Letter 0]

15 Dec 2023

PONE-D-23-33702A scoping review on the use of traditional medicine and oral health in AfricaPLOS ONE

Dear Dr. Folayan,

Thank you for submitting your manuscript to PLOS ONE. After careful consideration, we feel that it has merit but does not fully meet PLOS ONE’s publication criteria as it currently stands. Therefore, we invite you to submit a revised version of the manuscript that addresses the points raised during the review process.

We look forward to receiving your revised manuscript.

Kind regards,

Delfina Fernandes Hlashwayo, Ph.D.

Academic Editor

PLOS ONE

Journal Requirements:

**Additional Editor Comments:**

In addition to the reviewers' comments, please consider the following:

**A. Formatting and Style**

Please ensure that the manuscript adheres to the journal's guidelines, including font size for titles.Consider incorporating continuous line numbers to facilitate the identification of corrections.You have included sections on abbreviations and declarations. However, the journal's guidelines do not specifically call for these sections.

**B. Abstract**

The statement about the surveys identifying 29 to 62 different species from 15 to 29 different families is unclear, as these numbers differ. The same discrepancy is present in the results section of the manuscript. Please specify the total number of different plant species identified.

**C. Methods**

Provide clarity regarding the term "link" in the research question.Ensure consistency in referring to the research question in the singular when applicable.Clearly state whether the research protocol has been registered.Please specify the version of EndNote used.Please review and improve the construction of the first sentence in the inclusion criteria.Please simplify the excessively long second sentence in the exclusion criteria.Clearly indicate the source used for the subdivision of Africa into geographical terms.

**D. Results:**

Please correct grammatical errors in the PRISMA Flow chart, specifically in the "Not Africa study".On page 6, the authors mention, "Table 1 shed light." Please choose a more refined expression.Consider presenting information about plant families, parts, and administration modes as a figure instead of text.Ensure that scientific names in Table 2 are properly italicized.Clarify the meaning of "0.93" in the third row of Table 3.

**E. Discussion:**

Please provide a more concrete discussion on the main findings of your study, such as the most cited plants, families, and promising plant activities. Address these aspects in both the overall discussion and the conclusion.

**F. Supplemental Materials**

Please include the dates of the research in the first supplemental material.Consider including a supplemental file containing details about the study, including the name of each plant, preparation method, administration methods and other relevant details. Another supplemental file should cover the experimental studies with detailed information, including the results of antibacterial activity.

Reviewers' comments:

Reviewer's Responses to Questions

**Comments to the Author**

1. Is the manuscript technically sound, and do the data support the conclusions?

Reviewer #1: Yes

Reviewer #2: Yes

Reviewer #3: Yes

2. Has the statistical analysis been performed appropriately and rigorously? 

Reviewer #1: N/A

Reviewer #2: N/A

Reviewer #3: Yes

3. Have the authors made all data underlying the findings in their manuscript fully available?

Reviewer #1: Yes

Reviewer #2: Yes

Reviewer #3: Yes

4. Is the manuscript presented in an intelligible fashion and written in standard English?

Reviewer #1: Yes

Reviewer #2: Yes

Reviewer #3: Yes

5. Review Comments to the Author

Reviewer #1: The manuscript reviewed publications that focused on the traditional usage of medicinal plants to treat oral diseases and oral hygiene to identify knowledge gaps. The authors found that, from a pool of more than 500 publications, only 12 studies focused on oral hygiene. Therefore, the authors concluded that there is a need for country-specific traditional medicinal studies in Africa. Although I agree with this statement, the authors should have included review studies as well to get a more accurate picture of how big or small the gap is. The number given here could be an underestimate. There are several ethnobotanical studies that make mention of oral diseases, which were later incorporated into review papers on oral hygiene.

Reviewer #2: The authors of this manuscript have written a comprehensive scoping review on the use of traditional medicine and oral health in Africa. They have explained quite well the objective of this review which was to to chart the landscape of literature concerning the precise applications of traditional medicine in managing specific oral diseases. The methodology to do so was well chosen and explained, mainly in what concerns the databases chosen for the article search, the criteria selected for inclusion or exclusion of the scientific articles found in the literature, and the use of the Joanna Briggs Institute guidance. Furthermore, considering that a scoping review should be conducted to examine the extent, range, and nature of available research on a topic or question, it is my believe that the authors followed that line of thought when elaborating this manuscript, which is very well written.

Reviewing the manuscript’s PDF is quite difficult as it does not have numbered lines. To help with the review process, I have highlighted some spelling errors on the PDF that I have attached.

Reviewer #3: According to the objectives, In the methodology section, the articles that are also clinical trials should be considered and the results should be written in the tables.

In the discussion section, to make the article more fruitful, it is better to point out the effective components that are found in more plant species and to investigate the relationship between oral and dental diseases with them.

6. PLOS authors have the option to publish the peer review history of their article (what does this mean?). If published, this will include your full peer review and any attached files.

Reviewer #1: **Yes: **Eunezia Sitoe

Reviewer #2: **Yes: **Catarina Milho

Reviewer #3: No

---

## [Author Response · Author response to Decision Letter 0]

16 Dec 2023

PONE-D-23-33702

A scoping review on the use of traditional medicine and oral health in Africa

Date of revision: 16th December 2023

We would like to thank the reviewers for their comprehensive review. This has helped improve the quality of our manuscript. Please find below a point by point response to the comments of the reviewers and the editor.

Additional Editor Comments:

In addition to the reviewers' comments, please consider the following:

A. Formatting and Style

1. Please ensure that the manuscript adheres to the journal's guidelines, including font size for titles.

Response: Thanks for this highlight. We have revised the manuscript to adhere to the journal guidelines.

2. Consider incorporating continuous line numbers to facilitate the identification of corrections.

Response: the continuous line numbers have been incorporated.

3. You have included sections on abbreviations and declarations. However, the journal's guidelines do not specifically call for these sections.

Response: This has now been deleted. 

B. Abstract

The statement about the surveys identifying 29 to 62 different species from 15 to 29 different families is unclear, as these numbers differ. The same discrepancy is present in the results section of the manuscript. Please specify the total number of different plant species identified.

Response: we edited the statement to read: The number of plant species identified by the seven ethnobotanical surveys ranged from 29 to 62 while the number of plan families ranged from 15 to 29.

C. Methods

1. Provide clarity regarding the term "link" in the research question.

Response: We edited the word and wrote: what are published about the use of traditional medicines for oral health in Africa.

2. Ensure consistency in referring to the research question in the singular when applicable.

Response: Thanks for raising this. We noted and addressed this edit in the manuscript.

3. Clearly state whether the research protocol has been registered.

Response: The protocol was not registered since it is not required that scoping review manuscripts be registered (https://libguides.uta.edu/ScopingReviews/protocol#:~:text=Unlike%20protocols%20for%20Systematic%20Reviews,many%20issues%20down%20the%20line!). 

4. Please specify the version of EndNote used.

Response: We have included this detail. EndNote 7.8

5. Please review and improve the construction of the first sentence in the inclusion criteria.

Response: We have corrected the statement and wrote: Peer-reviewed journal articles, books, book chapters, conference proceedings, and reports that cover traditional medicine and oral health focusing on Africa that addressed both preventive and curative aspects herbal remedies and oral health were included in the review.

6. Please simplify the excessively long second sentence in the exclusion criteria.

Response: Thanks for raising this. Sentence has now been edited. 

7. Clearly indicate the source used for the subdivision of Africa into geographical terms.

Response: Please find the link to the sub-division of Africa. This information has been provided in the manuscript. [https://www.uneca.org/subregional-offices]

D. Results:

1. Please correct grammatical errors in the PRISMA Flow chart, specifically in the "Not Africa study".

Response: Done

2. On page 6, the authors mention, "Table 1 shed light." Please choose a more refined expression.

Response: We edited and wrote: Table 1 highlights the traditional medicines used to manage various oral diseases and conditions.

3. Consider presenting information about plant families, parts, and administration modes as a figure instead of text.

Response: Thanks for the suggestion. The team could not identify the best way to do this presentation other than in the table format and so we have left it in the current form.

4. Ensure that scientific names in Table 2 are properly italicized.

Response: Thanks for this suggestion. Done. We also received guidance on this from one of the peer reviewers. 

5. Clarify the meaning of "0.93" in the third row of Table 3.

Response: It was the value for the information consensus factor. We have now deleted the figure. 

E. Discussion:

Please provide a more concrete discussion on the main findings of your study, such as the most cited plants, families, and promising plant activities. Address these aspects in both the overall discussion and the conclusion.

Response: we wrote: The most prominent plant family identified for the management of oral diseases was Solanaceae. It has a widespread distribution but particularly concentrated in South America [36]. The family's most economically significant genus in Africa is Solanum notably Solanum tuberosum (potato), Solanum lycopersicum (tomato), and Solanum melongena (eggplant) [37]. In both rural and urban India, various parts of 17 Solanaceae species, such as fruits, seeds, berries, flowers, twigs, leaves, and roots, as well as the whole plant, have been traditionally employed for dental care to manage toothache [38]. Solanaceae is rich in biologically active alkaloids and exhibits antioxidant and anti-inflammatory properties [39]. When utilized as an adjunctive antiseptic mouth rinse by individuals adhering to recommended oral hygiene practices like brushing and flossing, it effectively reduces the risk of periodontal diseases [40].

Solanaceae and several other plant families are of importance for oral health because of their antimicrobial properties [41]. Studies on the use of these natural plants for oral health are being conducted in other countries outside of Africa such as Australia, Brazil, France, Iran, Japan, Mexico, Switzerland, and the United Kingdom. However, there is a notable scarcity of studies examining the clinical efficacy and toxicity of these products for oral health within Africa: several plants recommended for oral disease treatment may possess inherent dangers and toxicity [23]. In contrast to other regions globally where research explores incorporating these natural products into toothpaste formulations, African countries currently prioritize bioprospecting and primary production of the raw plants known for their antimicrobial properties [41].

F. Supplemental Materials

1. Please include the dates of the research in the first supplemental material.

Response: Thanks a million. We do not have clarity about this suggested edits. We have only one supplementary material and it was information about the search strategies. The three tables contain the dates of the research. Please do provide further clarification on the edit required. 

2. Consider including a supplemental file containing details about the study, including the name of each plant, preparation method, administration methods and other relevant details. Another supplemental file should cover the experimental studies with detailed information, including the results of antibacterial activity.

Response: Thanks for this suggestion. However, these details analysis are already contained in the main manuscripts of most of the journals reviewed. While we appreciate this listing, we collectively felt such listing would not be providing a new perspective to the field as the listing are already done. Such additional listing will be viewed as compendium which is outside the scope of the current study. With due respect to the perspective of the editor, we felt we will like to live this compendium out of this piece of work.

Reviewer #1: The manuscript reviewed publications that focused on the traditional usage of medicinal plants to treat oral diseases and oral hygiene to identify knowledge gaps. The authors found that, from a pool of more than 500 publications, only 12 studies focused on oral hygiene. Therefore, the authors concluded that there is a need for country-specific traditional medicinal studies in Africa. Although I agree with this statement, the authors should have included review studies as well to get a more accurate picture of how big or small the gap is. The number given here could be an underestimate. There are several ethnobotanical studies that make mention of oral diseases, which were later incorporated into review papers on oral hygiene.

Response: We recognize the study's limitations by stating: However, it is essential to acknowledge certain constraints in this study. The primary focus is on publications available in English and accessible through the specific database employed for this research. It is pertinent to mention that research publications in Africa are presented in various languages, including French and German. Nevertheless, to address the study objective, we conducted an extensive search of English literature, and the comprehensive findings of this search are reported in this review. While we concur with the authors regarding the abundance of ethnobotanical studies, our focus is specifically on those addressing oral diseases and published in English.

Reviewer #2: The authors of this manuscript have written a comprehensive scoping review on the use of traditional medicine and oral health in Africa. They have explained quite well the objective of this review which was to to chart the landscape of literature concerning the precise applications of traditional medicine in managing specific oral diseases. The methodology to do so was well chosen and explained, mainly in what concerns the databases chosen for the article search, the criteria selected for inclusion or exclusion of the scientific articles found in the literature, and the use of the Joanna Briggs Institute guidance. Furthermore, considering that a scoping review should be conducted to examine the extent, range, and nature of available research on a topic or question, it is my believe that the authors followed that line of thought when elaborating this manuscript, which is very well written. Reviewing the manuscript’s PDF is quite difficult as it does not have numbered lines. To help with the review process, I have highlighted some spelling errors on the PDF that I have attached.

Response: we thank the reviewer for the feedback. We accessed the suggested edits in the PDF and have effected all suggested corrections. 

Reviewer #3: According to the objectives, In the methodology section, the articles that are also clinical trials should be considered and the results should be written in the tables.

Response: We identified experimental studies. We did not identify clinical trials. One of the gaps in the literatures on the use of herbal remedies for oral health in Africa is the lack of clinical trials and toxicology studies. Prior studies had recognised this gap in research [23, 41] and had recommended increased funding for these studies in Africa. 

In the discussion section, to make the article more fruitful, it is better to point out the effective components that are found in more plant species and to investigate the relationship between oral and dental diseases with them.

Response: thanks for raising this. We have expanded on this discussion as highlighted and we have written: The most prominent plant family identified for the management of oral diseases was Solanaceae. It has a widespread distribution but particularly concentrated in South America [36]. The family's most economically significant genus in Africa is Solanum notably Solanum tuberosum (potato), Solanum lycopersicum (tomato), and Solanum melongena (eggplant) [37]. In both rural and urban India, various parts of 17 Solanaceae species, such as fruits, seeds, berries, flowers, twigs, leaves, and roots, as well as the whole plant, have been traditionally employed for dental care to manage toothache [38]. Solanaceae is rich in biologically active alkaloids and exhibits antioxidant and anti-inflammatory properties [39]. When utilized as an adjunctive antiseptic mouth rinse by individuals adhering to recommended oral hygiene practices like brushing and flossing, it effectively reduces the risk of periodontal diseases [40].

Solanaceae and several other plant families are of importance for oral health because of their antimicrobial properties [41]. Studies on the use of these natural plants for oral health are being conducted in other countries outside of Africa such as Australia, Brazil, France, Iran, Japan, Mexico, Switzerland, and the United Kingdom. However, there is a notable scarcity of studies examining the clinical efficacy and toxicity of these products for oral health within Africa: several plants recommended for oral disease treatment may possess inherent dangers and toxicity [23]. In contrast to other regions globally where research explores incorporating these natural products into toothpaste formulations, African countries currently prioritize bioprospecting and primary production of the raw plants known for their antimicrobial properties [41].

---

## [Decision Letter · Decision Letter 1]

9 Jan 2024

A scoping review on the use of traditional medicine and oral health in Africa

PONE-D-23-33702R1

Dear Dr. Foláyan,

We’re pleased to inform you that your manuscript has been judged scientifically suitable for publication and will be formally accepted for publication once it meets all outstanding technical requirements.

Kind regards,

Faham Khamesipour, Ph.D.

Academic Editor

PLOS ONE

Additional Editor Comments (optional):

Reviewers' comments:

Reviewer's Responses to Questions

**Comments to the Author**

1. If the authors have adequately addressed your comments raised in a previous round of review and you feel that this manuscript is now acceptable for publication, you may indicate that here to bypass the “Comments to the Author” section, enter your conflict of interest statement in the “Confidential to Editor” section, and submit your "Accept" recommendation.

Reviewer #1: All comments have been addressed

Reviewer #3: All comments have been addressed

2. Is the manuscript technically sound, and do the data support the conclusions?

Reviewer #1: Yes

Reviewer #3: Yes

3. Has the statistical analysis been performed appropriately and rigorously? 

Reviewer #1: N/A

Reviewer #3: Yes

4. Have the authors made all data underlying the findings in their manuscript fully available?

Reviewer #1: Yes

Reviewer #3: Yes

5. Is the manuscript presented in an intelligible fashion and written in standard English?

Reviewer #1: Yes

Reviewer #3: Yes

6. Review Comments to the Author

Reviewer #1: All suggestions made by the editor and other reviewers were taken into consideration by the authors. The paper might be helpful for upcoming studies.

Reviewer #3: (No Response)

7. PLOS authors have the option to publish the peer review history of their article (what does this mean?). If published, this will include your full peer review and any attached files.

Reviewer #1: No

Reviewer #3: **Yes: **Zarrin Sarhadynejad
